# Duplex On-Site Detection of *Vibrio cholerae* and *Vibrio vulnificus* by Recombinase Polymerase Amplification and Three-Segment Lateral Flow Strips

**DOI:** 10.3390/bios11050151

**Published:** 2021-05-12

**Authors:** Pei Wang, Lei Liao, Chao Ma, Xue Zhang, Junwei Yu, Longyu Yi, Xin Liu, Hui Shen, Song Gao, Qunwei Lu

**Affiliations:** 1Key Laboratory of Molecular Biophysics of Ministry of Education, Department of Biomedical Engineering, College of Life Science and Technology, Center for Human Genome Research, Huazhong University of Science and Technology, Wuhan 430074, China; wangpei@hust.edu.cn (P.W.); ylyhust@hust.edu.cn (L.Y.); xliu@mail.hust.edu.cn (X.L.); 2Jiangsu Key Laboratory of Marine Pharmaceutical Compound Screening, Jiangsu Key Laboratory of Marine Biological Resources and Environment, Co-Innovation Center of Jiangsu Marine Bio-Industry Technology, School of Pharmacy, Jiangsu Ocean University, Lianyungang 222005, China; 2019220321@jou.edu.cn (L.L.); hhitmachao@163.com (C.M.); 2020220673@jou.edu.cn (X.Z.); 3Ustar Biotechnologies (Hangzhou) Ltd., Hangzhou 310053, China; junwei.yu@bioustar.com; 4Jiangsu Institute of Oceanology and Marine Fisheries, Nantong 226007, China; darkhui@163.com

**Keywords:** *Vibrio cholerae*, *Vibrio vulnificus*, recombinase polymerase amplification, lateral flow strip, on-site detection, multiplexing

## Abstract

*Vibrio cholerae* and *Vibrio vulnificus* are two most reported foodborne *Vibrio* pathogens related to seafood. Due to global ocean warming and an increase in seafood consumption worldwide, foodborne illnesses related to infection of these two bacteria are growing, leading to food safety issues and economic consequences. Molecular detection methods targeting species-specific genes are effective tools in the fight against bacterial infections for food safety. In this study, a duplex detection biosensor based on isothermal recombinase polymerase amplification (RPA) and a three-segment lateral flow strip (LFS) has been established. The biosensor used *lolB* gene of *Vibrio cholerae* and *empV* gene of *Vibrio vulnificus* as the detection markers based on previous reports. A duplex RPA reaction for both targets were constructed, and two chemical labels, FITC and DIG, of the amplification products were carefully tested for effective and accurate visualization on the strip. The biosensor demonstrated good specificity and achieved a sensitivity of 10^1^ copies per reaction or one colony forming unit (CFU)/10 g of spiked food for both bacteria. Validation with clinical samples showed results consistent with that of real-time polymerase chain reaction. The detection process was simple and fast with a 30-min reaction at 37 °C and visualization on the strip within 5 min. With little dependence on laboratory settings, this biosensor was suitable for on-site detection, and the duplex system enabled simultaneous detection of the two important foodborne bacteria. Moreover, the principle can be extended to healthcare and food safety applications for other pathogens.

## 1. Introduction

The ubiquitous, Gram-negative halophilic bacteria of the *Vibrio* genus are capable of infecting animals including humans [1]. Many of them are related to marine animals such as shrimp, fish, and shellfish, and have posted threats to public health by causing foodborne illnesses through contamination of seafood [2]. The rising seawater temperature in recent years has benefited the growth of *Vibrio* species, meanwhile the seafood consumption is increasing worldwide [3]. These factors have heightened the concern of *Vibrio* pathogens on food safety. *Vibrio cholerae* and *Vibrio vulnificus* are two of those most reported foodborne *Vibrio* pathogens related to seafood [4]. Accurate and reliable detection methods of them are important for both food safety controls and epidemiological monitoring.

Molecular detection methods targeting species-specific genes are effective tools in fighting against bacterial infections for food safety [5]. These methods are based on DNA amplification principles, including polymerase chain reaction (PCR), real-time PCR (qPCR), loop-mediated isothermal amplification (LAMP) [6], and recombinase polymerase amplification (RPA) [7]. To give some examples, PCR- and qPCR- based methods targeting genes *rpoA* (genus-specific RNA polymerase subunit A), *ctxA* (cholera toxin subunit A), *toxR* (global regulatory gene), etc. have been developed for the detection of *V. cholerae* [8,9]. Genes *vvhA* (cytolysin), *pilF* (pilus-type IV assembly protein), *hly* (hemolysin), etc. have been selected as the targets of PCR- and qPCR- based detection methods for *V. vulnificus* [4,10,11]. To reduce the dependence on sophisticated thermocyclers for on-site detections, LAMP-based methods have been developed targeting the *ompW* (outer membrane protein) gene of *V. cholerae*, and *rpoS* (RNA polymerase subunit sigma factor S) and *vvhA* genes of *V. vulnificus* [12,13,14]. RPA reactions can amplify DNA targets isothermally under relatively low temperatures and are more convenient for on-site detections than LAMP [15,16]. For more on-site detection choices, RPA assays targeting *lolB* (outer membrane lipoprotein) gene of *V. cholerae* and *empV* (extracellular metalloproteinase) gene of *V. vulnificus* have been developed [17,18]. These molecular detection methods have improved the rapidity, sensitivity, and accuracy of the detection practice of these two important *Vibrio* pathogens, and have made good contributions to food safety and public health.

Because identifying various pathogenic *Vibrio* species in food contaminations is a practical requirement, multiplex approaches have been made with the aim of targeting multiple species in a single assay. These were mainly qPCR approaches, TaqMan or SYBR Green-based, with which multiple *Vibrio* species, including *V. cholerae* and *V. vulnificus,* could be simultaneously detected in a single run [4,19]. To date, isothermal amplification-based multiplex detection assays targeting multiple *Vibrio* species have not been reported.

In this study, a duplex detection biosensor for *V. cholerae* and *V. vulnificus* based on RPA and a 3-segment lateral flow strip (LFS) has been established. The biosensor had a good specificity and achieved a sensitivity of 10^1^ copies per reaction or 1 colony forming unit (CFU)/10 g of spiked food for both bacteria. The detection included a 30-min reaction at 37 °C and visualization on the strip within 5 min. The biosensor was suitable for on-site detection, and the duplex system enabled simultaneous detection of the two important foodborne bacteria, which provided more convenience. In addition, the principle can be extended to healthcare and food safety applications for other pathogens.

## 2. Materials and Methods

### 2.1. Bacterial Strains

The 16 *Vibrio* species reference strains including *Vibrio cholerae*, *Vibrio vulnificus*, *Vibrio parahemolyticus*, *Vibrio alginolyticus*, *Vibrio harveyi*, *Vibrio mediterranei*, *Vibrio shilonii*, *Vibrio splendidus*, *Vibrio mimicus*, *Vibrio ichthyoenteri*, *Vibrio campbellii*, *Vibrio chagasii*, *Vibrio fluvialis*, *Vibrio natriegens*, *Vibrio ponticus* and *Vibrio rotiferianus* were obtained from the Jiangsu Institute of Oceanology and Marine Fisheries (Nantong, China). The 3 common foodborne pathogenic species reference strains *Salmonella* Typhimurium, *Listeria monocytogenes*, and *Staphylococcus aureus* were purchased from American Type Culture Collection, Manassas, VA, USA. Among these species, *V. cholerae* and *V. vulnificus* were the detection targets of this study, and other species were included as controls. All strains were confirmed by 16S rRNA sequencing [20]. Information of the strains are listed in Table 1. For template preparation, genomic DNA of the bacterial strains were extracted with the TIANamp Bacteria DNA Kit (Tiangen Biotech Co Ltd., Beijing, China) and quantified by a Qubit 4 fluorometer (Thermo Fisher Scientific Inc., Wilmington, DE, USA).

### 2.2. Primers and Probes

The primer and probe sequences used in this study were derived from previously reported RPA-based individual assays for *V. cholerae* and *V. vulnificus* detections (Table 2) [17,18]. The target genes for *V. cholerae* and *V. vulnificus* were the outer membrane lipoprotein gene *lolB* (GenBank accession number AP014524.1) and the metalloprotease gene *empV* (GenBank accession number U50548.1), respectively. The primer and probe design followed the manufacturer’s instructions of the RAA-nfo Nucleic Acid Amplification Reagent (Hangzhou ZC Bio-Sci & Tech Co Ltd., Hangzhou, China). Chemical modifications [fluorescein isothiocyanate (FITC), digoxigenin (DIG) or biotin] were made to the 5′ ends of the probes and the reverse primers to facilitate differential visualization on the strips. For each probe, a base in the middle was replaced with a tetrahydrofuran (THF), and the 3′ end was blocked with a Spacer C3 (SpC3). This probe design could reduce primer-dependent artifacts when using the RAA-nfo Nucleic Acid Amplification Reagent [7]. All the primers and probes were synthesized by General Biosystems Co Ltd., Anhui, China.

### 2.3. RPA-LFS

RPA reactions were performed according to the manufacturer’s instructions of the RAA-nfo Nucleic Acid Amplification Reagent (Hangzhou ZC Bio-Sci & Tech Co Ltd.). The reactions were performed at 37 °C for 30 min. After the reaction, 5 μL of the reaction mixture was applied to the sample pad of the LFS, and the starting end of the LFS was put into 100 μL of the strip solvent (Ustar Biotechnologies Ltd., Hangzhou, China). The test and control lines were visualized in 5 min.

### 2.4. Preparation of Spiked Food Samples

Spiked food samples were used to determine the sensitivity of the biosensor in food matrix. Shrimp was selected as the representative food sample. After purchased from a local supermarket, the shrimp samples were confirmed to have no contamination of *V. cholerae* or *V. vulnificus* using qPCR [21,22]. Homogenization of the shrimp samples were conducted with the handheld 3rd Gen. TGrinder (Tiangen Biotech Co Ltd.). To 100 mL of alkaline peptone broth (Sinopharm Chemical Reagent Co Ltd., Beijing, China), 10 g of the homogenate was added and suspended thoroughly, and the suspension was spiked with 10^0^, 10^1^, or 10^2^ CFU of the mix of *V. cholerae* and *V. vulnificus*. For enrichment, the suspension was put in a shaker and incubated with 200 rpm shaking under 30 °C. At each planned time points, 1 mL of the enrichment suspension was collected and centrifuged at 5000× *g* for 5 min. The pelleted bacterial cells were resuspended with 200 µL of water. DNA was released by boiling at 100 °C for 10 min, and 1 µL of the boiled resuspension was used as the template for RPA-LFS.

### 2.5. Clinical Samples

Clinical samples were used to validate the biosensor. The clinical samples were kindly provided by the Jiangsu Institute of Oceanology and Marine Fisheries (Nantong, China). There were 20 shrimp samples (*Litopenaeus vannamei*, sampling tissue: hepatopancreas), 10 fish samples (*Carassius auratus*, sampling tissue: liver), and 10 shellfish samples (*ostrea gigas thunberg*, sampling tissue: meat). The samples were disinfected with ethanol, and 1 g of each sample was put into 9 mL of PBS and homogenized by the handheld grinder. One milliliter of the sample homogenate was boiled at 100 °C for 10 min, and centrifuged at 8000× *g* for 2 min. The supernatant (1 µL) was used as the template for RPA-LFS or qPCR.

### 2.6. qPCR

Specific primers (Table 2) described in previous reports for the detections of *V. cholerae* and *V. vulnificus* were used in qPCR assays of this study as reference methods [21,22]. The targeted genes were *lolB* gene of *V. cholerae* and *gyrB* (DNA gyrase subunit B) gene of *V. vulnificus*. The qPCR reaction mixtures (20 μL) contained 1 μl of the DNA template, 0.4 μL of each primer (10 μM), and 10 μL of the 2X SYBR Green qPCR Mix (Vazyme Biotech Co Ltd., Nanjing, China). The cycling programs followed the reported procedures [21,22]. The qPCR assays were performed on an Applied Biosystems 7900HT Fast Real-Time PCR System. A cycle threshold (*Ct*) less than 32 was considered positive.

## 3. Results

### 3.1. Principle of the Biosensor

The principle of the duplex detection biosensor for *V. cholerae* and *V. vulnificus* was based on duplex RPA amplification of the target gene fragments followed by visualization of the amplification products on a three-segment LFS (Figure 1). The *lolB* gene of *V. cholerae* and *empV* gene of *V. vulnificus* had been used as molecular markers for detection of these two bacteria in previously reported assays [17,18]. In this study, these target gene fragments were specifically amplified in RPA reactions with chemically labeled primers and probes (Figure 1a). The use of FITC- or DIG- labeled probes and biotin-labeled reverse primers could generate amplification products with FITC or DIG at one end and biotin at the other with the RAA-nfo Nucleic Acid Amplification Reagent (Hangzhou ZC Bio-Sci & Tech Co Ltd.). Efficient one-tube duplex amplification for the two detection markers was achieved by reaction optimization (Figure 1b), and the amplification products were applied to a three-segment LFS for simultaneous detection of the two bacteria (Figure 1c). The strip had the sample pad, the conjugate pad, the three detection segments and the absorbent pad on the solvent migration route. The conjugate pad was soaked with streptavidin-bound gold nanoparticles (streptavidin-AuNPs). The three detection segments were DIG line (anti-DIG antibody coated), FITC line (anti-FITC antibody coated) and C line (biotin coated control line). Amplification products of the *lolB* gene fragment of *V. cholerae* had the DIG labeling and should be visualized at the DIG line, while amplification products of the *empV* gene fragment of *V. vulnificus* had the FITC labeling and should be visualized at the FITC line. Visualization of the C line should indicate the proper completion of the strip assay. Selection of FITC labeling for *V. vulnificus* and DIG labeling for *V. cholerae*, and the order of DIG-, FITC-, and C-lines on the solvent migration route of the strip, were determined in this study (see below).

### 3.2. Optimization of the Duplex RPA Reaction

In the RPA reaction, chemical labels of the probes remain on one end of the amplification products (Figure 1a). FITC and DIG are chemical groups usually used for 5′-end labeling of nucleic acids [23]. In this study, they were selected as the labeling groups of the probes to facilitate visualization of the RPA amplification products on the strip through antibody-antigen interactions (Figure 1c). We designed primers and probes according to previously established RPA detection assays [17,18] (Table 2), and selected FITC labeling for *V. vulnificus* (probe VV-P1) and DIG labeling for *V. cholerae* (probe VC-P1) to start with.

To build the duplex RPA reaction, the optimal concentration range of the primers and probes in the single reaction was firstly determined. Different template amounts were used to assess the limit of detection (LOD) of the single reactions for *V. vulnificus* and *V. cholerae*, and the LOD of 10^1^ gene copies per reaction was observed for both bacteria (Figure 2a,b, left panels). Then the concentrations of the primers and probes were decreased proportionally to determine the lowest primer/probe concentrations that could be used (Figure 2a,b, right panels). The results showed that, in the single reactions for both bacteria, the primer/probe concentrations at 50% of what the manufacturer recommended ((primers) = 160 nM; (probe) = 48 nM) had no obvious effect on the signal intensity. Thus, we determined to use 160 nM of each primer and 48 nM of each probe in the duplex RPA reaction. Validation of the duplex reaction showed unaffected LOD comparing to the single reactions (Figure 2c).

### 3.3. Determination of Visualization Lines on the 3-Segment Strip

The amplicons from the duplex reactions were applied to the 3-segment strips. In the first 3-segment design of the strip, the 3 lines on the solvent migration route were FITC-, DIG-, and C-lines. The initial amplicon labeling design was FITC labeling for *V. vulnificus* and DIG labeling for *V. cholerae* (probe combination 1: probes VC-P1 and VV-P1 used). Surprisingly, the visualization on the DIG line was very weak (Figure 3a). Then we tried to change the amplicon labeling to DIG for *V. vulnificus* and FITC for *V. cholerae* (probe combination 2: probes VC-P2 and VV-P2 used) (Table 2). However, the visualization on the DIG line did not improve (Figure 3b). Then, another three-segment design of the strip was used, of which the three lines on the solvent migration route were DIG-, FITC-, and C-lines. With this three-segment design, amplicons with the probe combination 1 visualized normally, while amplicons with the probe combination 2 had weaker visualizations (Figure 3c,d). Thus, the three-segment design of the strip should be DIG-, FITC-, and C-lines in order on the solvent migration route, and the amplicon labeling should be FITC for *V. vulnificus* and DIG for *V. cholerae* (probe combination 1).

### 3.4. Specificity and Sensitivity

The specificity of this biosensor was established with a series of *Vibrio* species and several commonly seen foodborne pathogens. The results showed that *V. vulnificus* and *V. cholerae* templates gave positive signals at the corresponding lines, and all other templates gave negative signals (Figure 4a). This suggested good specificity of the biosensor.

The detection sensitivity was tested with serial dilutions of the bacteria DNAs. 10^0^–10^4^ copies of the genomic DNAs of *V. vulnificus* and *V. cholerae* were mixed and served as the template for the duplex amplification. The amplicons were applied to the 3-segment strip and showed a sensitivity of 10^1^ copies for both bacteria (Figure 4b). Additionally, the detection sensitivity was tested in food matrix. 10^0^–10^2^ CFU of *V. vulnificus* and *V. cholerae* were spiked in 10 g of shrimp meat, and the spiked samples were enriched for 0-8 hr followed by sample processing and duplex amplification. Application of the amplicons on the strips showed that as low as 1 CFU/10 g of *V. vulnificus* and *V. cholerae* could be detected after 4 h of enrichment (Figure 5). Thus, the sensitivity of the method was 10^1^ copies per reaction or 1 CFU/10 g of spiked food for both *V. vulnificus* and *V. cholerae*.

### 3.5. Clinical Sample Applications

A total of 40 clinical samples including shrimp (20 samples), fish (10 samples) and shellfish (10 samples) were collected from different areas of China and tested for *V. cholerae* and *V. vulnificus* by this duplex detection biosensor. The detection results were compared with qPCR (Table 3). Three shrimp samples, two fish samples and two shellfish samples were positive for *V. cholerae*. Three shrimp samples and one shellfish sample were positive for *V. vulnificus*. There was one sample positive for both bacteria. The detection results of the biosensor were consistent with that of qPCR.

## 4. Discussion

Detection of pathogenic *Vibrio* species in seafood contaminations is important for food safety and public health [2,24]. For the two important *Vibrio* pathogens, *V. cholerae* and *V. vulnificus*, molecular detection methods based on PCR, qPCR, LAMP, and RPA have been developed [8,9,12,13,17,18,25,26]. LAMP and RPA using isothermal amplification principles have provided greater convenience for on-site detections [6,7]. Efforts for multiplexing have been made using PCR and qPCR methods to detect multiple species in a single assay [8,9,25,26]. This study developed a duplex detection biosensor for *V. cholerae* and *V. vulnificus* based on RPA and a three-segment LFS, giving the on-site detection practice a more convenient choice.

To construct the duplex RPA reaction, one potential problem was the amplification preference (or bias) that could lead to nonuniformity of the amplicons [16,27]. This was essential because biased amplification of one target would affect the sensitivity of the reaction to the other. It has been reported that the amplification preference of RPA reactions could come from the size-dependent amplification rate of different targets [16,28]. In this study, we selected amplification targets with similar amplicon sizes for the two *Vibrio* species to prevent biased amplification in the duplex RPA reaction (Table 2). Moreover, the primers and probes were designed based on previously reported detection assays in which the specificity had been established [17,18], to prevent cross amplifications in the duplex reaction. Specifically, the primer/probe set for *V. cholerae* targeted the *lolB* gene, which was highly conserved in all serogroups of *V. cholerae*, but not conserved in other enteric bacterial species [29]. The primer/probe set for *V. vulnificus* targeted the *empV* gene, which also showed good specificity and evolutionary conservation [18].

We reduced the primer and probe concentrations to a point that the amplification efficiencies were not affected in reactions with single primer/probe sets, and then combined the two primer/probe sets at 1:1 molar ratio in one reaction mixture. The sensitivity was not affected in the duplex reaction when tested with double templates (*V. cholerae* and *V. vulnificus*) (Figure 2).

Surprisingly, when we applied the amplification products of the duplex RPA reaction to a three-segment strip on which the test lines were ordered as FITC-DIG-control, the visualization on the DIG line was very weak. This should not be a result of biased amplification, because when we exchanged the chemical labels of the amplicons (probe combination 1 changed to probe combination 2), the visualization of the test lines was the same. Moreover, there was no sign of biased amplification when using the 2-segment strips. Good visualization was obtained when a three-segment strip with the test lines ordered as DIG-FITC-control was used. The results indicated that the two test lines on the strip had to be placed in a particular order. A possible reason for this could be that there was a cross reaction between the DIG labeling on the amplicon and the anti-FITC antibody on the test line. The specificity of antibody–antigen reactions is usually considered high, but not to the extent of complete elimination of non-specific interactions [30]. For the design of three-segment strip in this study, it was important to determine the visualization lines by experimental testing.

After the duplex RPA reaction was built and the three-segment strip was determined, the duplex detection biosensor was established. This was for the first time an RPA-LFS-based multiplex detection assay targeting multiple *Vibrio* species was developed. Only a few similar assays had been developed for multiplex detection of microbial species, such as *Giardia*/*Cryptosporidium*/*Entamoeba* triplex detection [31], and *Streptococcus pneumoniae*/*Legionella pneumophila* duplex detection [32], as examples. Multiplexing with single-tube RPA is challenging because of the amplification bias as mentioned above, possibly caused by competition for the recombinase proteins from primers [16,27]. Careful optimizations of the RPA reaction thus are important for any multiplex assay. In addition, experimental testing for single-strip visualization of the amplicons is also essential as indicated by this study. Multiplexing on RPA-LFS detection requires sophisticated customization indeed.

The duplex biosensor from this study showed satisfactory performance. The specificity for *V. cholerae* and *V. vulnificus* was as good as the previously reported RPA-based individual assays [17,18]. The sensitivity was 10^1^ copies per reaction or 1 CFU/10 g of spiked food with enrichment for both the *Vibrio* pathogens. This sensitivity was also as good as the previously reported qPCR-, LAMP-, and RPA- based individual assays [12,13,17,18,21,33]. The biosensor could simultaneously detect two important foodborne *Vibrio* pathogens within 35 min (30 min of amplification and 5 min of visualization) under a convenient isothermal temperature of 37 °C. The template preparation was easy by sample boiling. Validation with clinical samples showed results consistent with that of qPCR. The simple and fast procedure made the duplex biosensor very suitable for on-site detection, providing a more convenient choice for the detection practice of *V. cholerae* and *V. vulnificus*. Moreover, the principle can be extended to healthcare and food safety applications for other pathogens.

## Figures and Tables

**Figure 1 biosensors-11-00151-f001:**
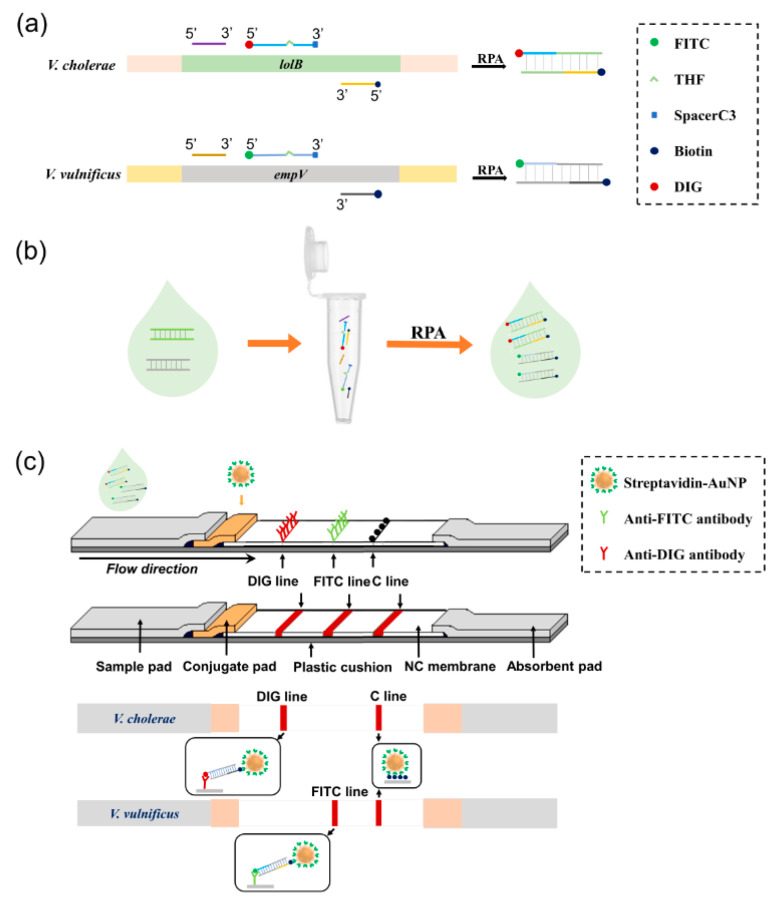
Schematic diagram of the duplex RPA-LFS biosensor. (**a**) Target amplification by RPA of the two *Vibrio* species. The target genes for *V. cholerae* and *V. vulnificus* were shown as horizontal strips. Primers and probes were shown as horizontal lines. Labels and modifications on the probes, the reverse primers and the RPA products were indicated. (**b**) Target amplification in the duplex RPA reaction. The duplex RPA system contained primer/probe sets for both detection markers. Both targets could be amplified with appropriate labels on the amplicons. (**c**) Simultaneous differential detection by visualization lines on the 3-segment strip: DIG, FITC and C (control) lines. The structure of the strip was shown with the coating materials indicated. RPA products from *V. cholerae* gave signal at the DIG line and that from *V. vulnificus* gave signal at the FITC line because of the different labels. Signal at the C line indicated proper finish of the lateral flow detection.

**Figure 2 biosensors-11-00151-f002:**
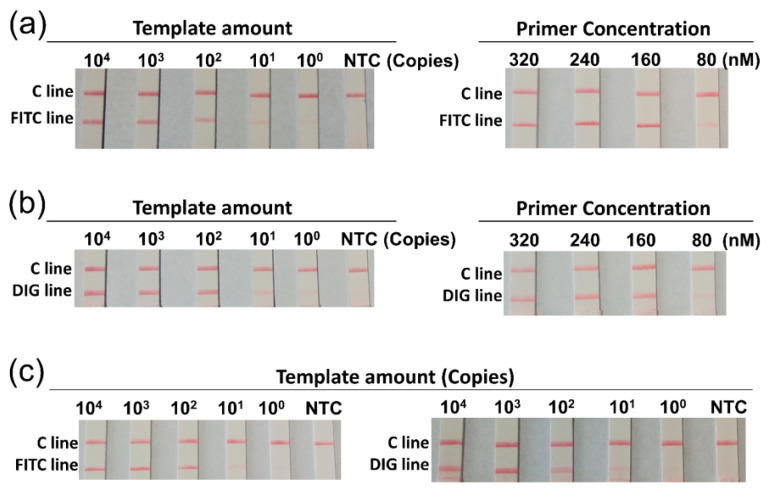
Optimization of the duplex RPA reaction. The concentration range of the primer/probe set in the single detection assay was determined for *V. vulnificus* (**a**) or *V. cholerae* (**b**). Different template amounts were tested in the single detection assays with the primer/probe concentrations set as recommended by the manufacturer of the reagent (the left panels of **a** and **b**). The final concentrations were: [forward primer] = 320 nM, [reverse primer] = 320 nM, and [probe] = 96 nM. The template amount was indicated on top of each strip. Using 10^2^ copies of the templates (the right panels of **a** and **b**), the primer/probe concentrations were decreased proportionally (forward primer:reverse primer:probe =1:1:0.3) to determine the concentration range. Concentration of each primer was indicated on top of the strip run amplicons of the corresponding reaction. Then the sensitivity of the duplex RPA reaction was confirmed (**c**). Mixed templates (double template of *V. vulnificus* and *V. cholerae*) of different amounts were tested and separately visualized for *V. vulnificus* (left panel of **c**) and *V. cholerae* (right panel of **c**). Final concentrations of the primers and probes in the duplex RPA reaction: [each primer] = 160 nM, and [each probe] = 48 nM. The NTC strips were the no-template controls. The 2-segment strips were used to visualize amplicons with corresponding labels. The images represent results from three independent experiments.

**Figure 3 biosensors-11-00151-f003:**
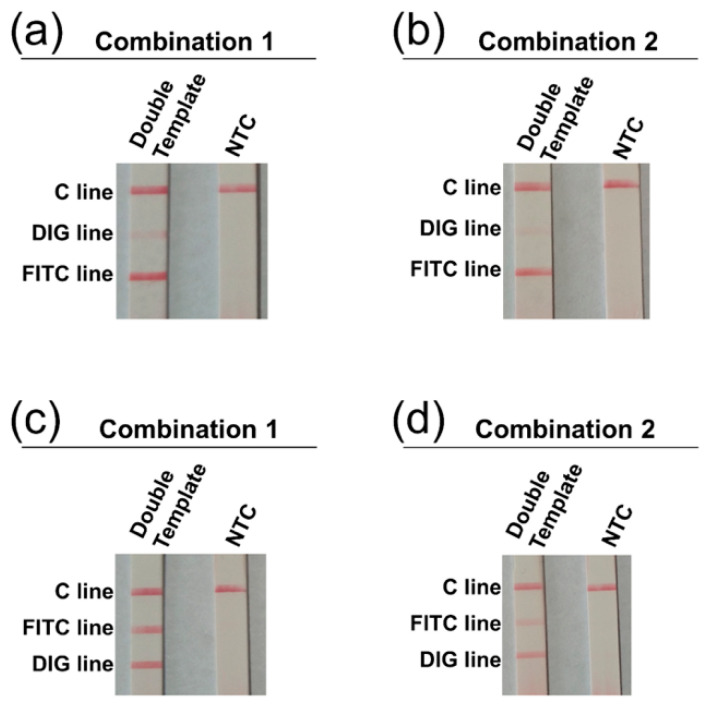
Determination of visualization lines on the 3-segment strip. Two orders of the visualization lines on the 3-segment strip were tested: FITC-DIG-C, and DIG-FITC-C. Two amplicon labeling designs were tested on each of the strip types: (1) FITC for *V. vulnificus* and DIG for *V. cholerae* (Combination 1: probes VC-P1 and VV-P1 used); (2) DIG for *V. vulnificus* and FITC for *V. cholerae* (Combination 2: probes VC-P2 and VV-P2 used). The duplex RPA products from probe Combination 1 and Combination 2 were visualized on the strips with the two orders of visualization lines: FITC-DIG-C (**a**,**b**), and DIG-FITC-C (**c**,**d**). “Double Template” meant the RPA reactions had both *V. vulnificus* and *V. cholerae* as the templates (10^4^ copies of each, mixed). The NTC strips were the no-template controls. The images represent results from three independent experiments.

**Figure 4 biosensors-11-00151-f004:**
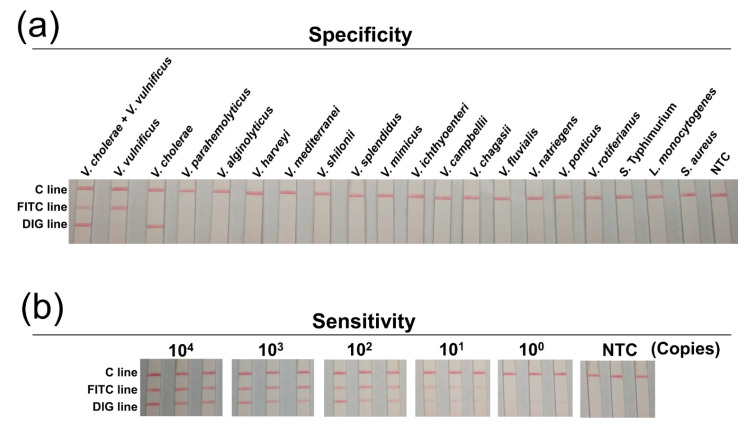
Specificity and sensitivity of the duplex RPA-LFS biosensor. (**a**) Specificity of the biosensor. Templates used for testing were indicated on top of each strip. The DNA concentrations of the templates were normalized to 10 ng/μL and 1 μL was used for each reaction. The image represented results from three independent experiments. (**b**) Sensitivity of the biosensor. Mixed templates (double template of *V. vulnificus* and *V. cholerae*) of different amounts were used for testing. The amount (in copies) was shown at the top of the strips. The triplicate strips represented results from three independent experiments. The NTC strips were the no-template controls.

**Figure 5 biosensors-11-00151-f005:**
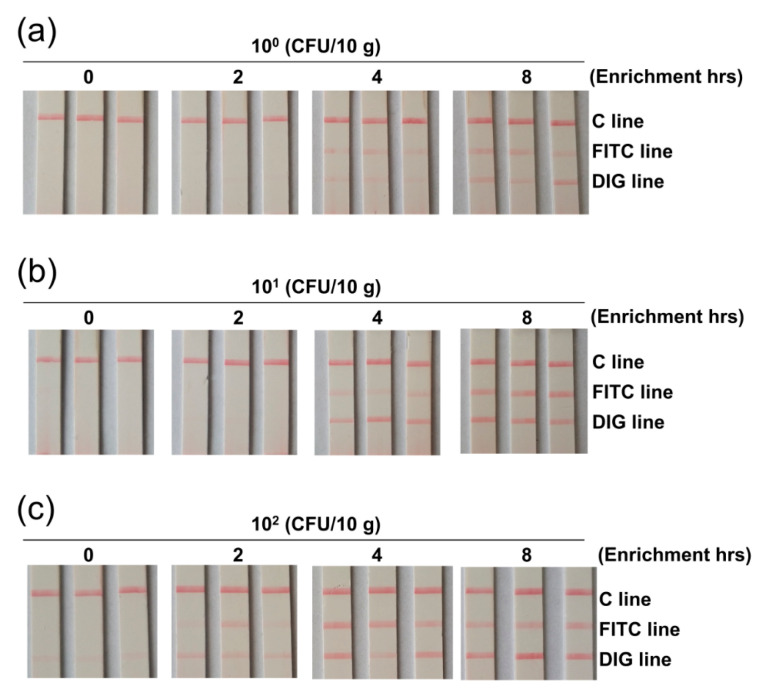
Limit of detection in spiked food samples. The image shows the detection results of the duplex RPA-LFS biosensor for spiked food samples. The spiking amounts were 10^0^ (**a**), 10^1^ (**b**), and 10^2^ (**c**) CFU of the mix of *V. cholerae* and *V. vulnificus* per 10 g of shrimp. The enrichment time (in hours) after spiking was indicated on top of the strips. The triplicate strips represented results from three independent experiments.

**Table 1 biosensors-11-00151-t001:** Bacteria strains used in this study.

Species	Strain Type	Designation
*Vibrio cholerae*	Reference strain	ATCC 14100
*Vibrio vulnificus*	Reference strain	ATCC 27562
*Vibrio parahemolyticus*	Reference strain	ATCC 17802
*Vibrio alginolyticus*	Reference strain	ATCC 17749
*Vibrio harveyi*	Reference strain	ATCC 43516
*Vibrio mediterranei*	Reference strain	ATCC 43341
*Vibrio shilonii*	Reference strain	ATCC BAA-91
*Vibrio splendidus*	Reference strain	MCCC 1A04096
*Vibrio mimicus*	Reference strain	MCCC 1A02602
*Vibrio ichthyoenteri*	Reference strain	MCCC 1A00057
*Vibrio campbellii*	Reference strain	MCCC 1A02605
*Vibrio chagasii*	Reference strain	MCCC 1B00386
*Vibrio fluvialis*	Reference strain	MCCC 1A02761
*Vibrio natriegens*	Reference strain	MCCC 1D00129
*Vibrio ponticus*	Reference strain	MCCC 1H00061
*Vibrio rotiferianus*	Reference strain	MCCC 1B00068
*Salmonella* Typhimurium	Reference strain	ATCC 14028
*Listeria monocytogenes*	Reference strain	ATCC 19115
*Staphylococcus aureus*	Reference strain	ATCC 6538

**Table 2 biosensors-11-00151-t002:** Primer and probe sequences.

Method	Target	Primer/Probe Name	Sequence (5′-3′)	Length (bp)	Amplicon Size (bp)
RPA	*V. cholerae*	VC-F	ATCTTCAAGCTGTTCAACGGGAATATCTAA	30	218
		VC-R	Biotin-ATCAGCGACAATCGTTCAACTTTCAATGGC	30	
		VC-P1	DIG-ATCAGGCTTTGTGCATCTTGGTCGCGGTAGA [THF] TTGATCATCATAAGTTTCG-SpC3	51	
		VC-P2	FITC-ATCAGGCTTTGTGCATCTTGGTCGCGGTAGA [THF] TTGATCATCATAAGTTTCG-SpC3	51	
	*V. vulnificus*	VV-F	GAGATGGATTCTTTGTATAACATTGCGT	28	214
		VV-R	Biotin-ACGATGACGTTGGTTGTGTTTCATTATC	28	
		VV-P1	FITC-GGTGAAGTTGGCTGGTGGTTATTTTCTGAA [THF] CATGGTTGTTGAGCTC-SpC3	47	
		VV-P2	DIG-GGTGAAGTTGGCTGGTGGTTATTTTCTGAA [THF] CATGGTTGTTGAGCTC-SpC3	47	
qPCR	*V. cholerae*	VC195F	CCGTTGAGGCGAGTTTGGTGAGA	23	195
		VC195R	GTGCGCGGGTCGAAACTTATGAT	23	
	*V. vulnificus*	gyr-vv1	GTCCGCAGTGGAATCCTTCA	20	285
		gyr-vv2	TGGTTCTTACGGTTACGGCC	20	

**Table 3 biosensors-11-00151-t003:** Detection of *V. cholerae* and *V. vulnificus* in clinical samples.

No.	Food Type	Sample Source	Detection Results for *V. cholerae*		Detection Results for *V. vulnificus*	
			RPA-LFS	qPCR	RPA-LFS	qPCR
1	Shrimp	Qingdao, China	-	-	-	-
2	Shrimp	Qingdao, China	-	-	-	-
3	Shrimp	Qingdao, China	-	-	-	-
4	Shrimp	Qingdao, China	-	-	-	-
5	Shrimp	Qingdao, China	+	+	-	-
6	Shrimp	Qingdao, China	-	-	-	-
7	Shrimp	Qingdao, China	-	-	+	+
8	Shrimp	Qingdao, China	+	+	+	+
9	Shrimp	Qingdao, China	-	-	-	-
10	Shrimp	Qingdao, China	-	-	-	-
11	Shrimp	Lianyungang, China	-	-	-	-
12	Shrimp	Lianyungang, China	-	-	-	-
13	Shrimp	Lianyungang, China	-	-	-	-
14	Shrimp	Lianyungang, China	-	-	-	-
15	Shrimp	Lianyungang, China	-	-	-	-
16	Shrimp	Lianyungang, China	-	-	+	+
17	Shrimp	Lianyungang, China	-	-	-	-
18	Shrimp	Lianyungang, China	+	+	-	-
19	Shrimp	Lianyungang, China	-	-	-	-
20	Shrimp	Lianyungang, China	-	-	-	-
21	Fish	Qingdao, China	-	-	-	-
22	Fish	Qingdao, China	-	-	-	-
23	Fish	Qingdao, China	-	-	-	-
24	Fish	Qingdao, China	-	-	-	-
25	Fish	Qingdao, China	-	-	-	-
26	Fish	Yancheng, China	+	+	-	-
27	Fish	Yancheng, China	+	+	-	-
28	Fish	Yancheng, China	-	-	-	-
29	Fish	Yancheng, China	-	-	-	-
30	Fish	Yancheng, China	-	-	-	-
31	Shellfish	Qingdao, China	-	-	-	-
32	Shellfish	Qingdao, China	+	+	-	-
33	Shellfish	Qingdao, China	-	-	-	-
34	Shellfish	Qingdao, China	+	+	-	-
35	Shellfish	Qingdao, China	-	-	-	-
36	Shellfish	Lianyungang, China	-	-	+	+
37	Shellfish	Lianyungang, China	-	-	-	-
38	Shellfish	Lianyungang, China	-	-	-	-
39	Shellfish	Lianyungang, China	-	-	-	-
40	Shellfish	Lianyungang, China	-	-	-	-

(+: positive result; -: negative result).

## Data Availability

The data presented in this study are available in the article, further inquiries can be directed to the corresponding author.

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
