# Peer review of "Duplex On-Site Detection of *Vibrio cholerae* and *Vibrio vulnificus* by Recombinase Polymerase Amplification and Three-Segment Lateral Flow Strips"

_biosensors, 2021, doi:10.3390/bios11050151_

Round 1
Reviewer 1 Report
This manuscript by Wang et al. reports on the development of a molecular approach for accurate and specific detection of Vibrio cholerae and Vibrio vulnificus, two bacterial species most associated with contaminated seafood. This assay showed high sensitivity and comparable results when compared to traditional Real-time PCR and can be successfully applied to other microorganisms.
Overall, the manuscript is well organized and written. It contains interesting findings that might be of great interest for Biosensor readers especially those working on developing molecular approaches for specific microbe detection. Some minor revisions are requested before acceptance for publication in Biosensor. Here are some specific edits that should be done to improve the current version of the manuscript:
- Has this developed system been used for the detection of other microbial species? If so, the authors should cite previous papers that has applied this system. If not, they should highlight more in the introduction the novelty of the developed approach.
- In paragraph 2.1 authors list all the bacterial species included in the study; they should precise which species are included as controls to avoid confusion.
- In Material and Methods: To make it less confusing for reader, the authors should state in each paragraph, the purpose and usage of each of the sea food samples (e.g. initial analyses of the duplex RPA-LFS biosensor., confirmation of the developed system, etc...)
- Authors used primer pairs from previous studies and showed high specificity towards Vibrio cholerae and Vibrio vulnificus. Were those primers designed on highly conserved regions? Or the gene targets are highly specific to Vibrio cholerae and Vibrio vulnificus. Please precise in the text.
Author Response
Thank you for your positive and constructive comments. Please see my point-by-point responses below. Any changes to the manuscript are highlighted in red color.
Point 1: Has this developed system been used for the detection of other microbial species? If so, the authors should cite previous papers that has applied this system. If not, they should highlight more in the introduction the novelty of the developed approach.
Response 1: This was for the first time an RPA-LFS-based multiplex detection assay targeting multiple Vibrio species was developed. Only a few similar assays had been developed for multiplex detection of microbial species. Multiplexing with single-tube RPA is challenging because of the amplification bias, and multiplexing on RPA-LFS detection requires sophisticated customization indeed. We feel it is more appropriate to added a paragraph in Discussion to convey this information. So, the 2nd last paragraph of the Discussion section has been added in the revised manuscript.
Point 2: In paragraph 2.1 authors list all the bacterial species included in the study; they should precise which species are included as controls to avoid confusion.
Response 2: A sentence has been added to this paragraph to clarify the target species and the controls.
Point 3: In Material and Methods: To make it less confusing for reader, the authors should state in each paragraph, the purpose and usage of each of the sea food samples (e.g. initial analyses of the duplex RPA-LFS biosensor., confirmation of the developed system, etc...)
Response 3: Sentences have been added to paragraphs 2.4 and 2.5 to state the purpose and usage of the seafood samples.
Point 4: Authors used primer pairs from previous studies and showed high specificity towards Vibrio cholerae and Vibrio vulnificus. Were those primers designed on highly conserved regions? Or the gene targets are highly specific to Vibrio cholerae and Vibrio vulnificus. Please precise in the text.
Response 4: The gene targets are highly specific to the respective species and are evolutionarily conserved. Information about the gene targets has been added to the 2nd paragraph of the Discussion.
Reviewer 2 Report
This manuscript developed a duplex RPA-biosensor for simultaneous detection of two Vibrio species from seafood. The authors clearly presented the different iterations of the biosensor and optimized the biosensor for optimal detection. Sensitivity and specificity were also determined by appropriate experiments. The study also included verification with spiked shrimp sample and the comparison of the biosensor and qPCR methods on real world seafood samples. Overall, the study was well designed and executed.
Only minor recommendations:
- Please clarify what exactly the 'double template' was in figure 3. How many copies of each bacterium?
- Still in figure 3, I find it not so easy to follow what combination 1 and combination 2 represent. The information was not included in figure 3 legend either but only in the paragraph from the previous page. Providing more information either directly in the figure or in the legend would make it much easier to read.
- In figures 4b and 5, only one image from three independent experiments was shown. Because these images were not quantitative data, lack of statistical analysis comprise the claims regarding detecion limit (such as the lowest detection limit was 1 copy). This could be addressed two ways: 1) display all three strips side by side to show that 1 copy or 1 cfu/10g shrimp can be consistently detected; or 2) using image analysis software to do densitometry analysis of the bands and quantitatively graph the results with mean +/- sd
Author Response
Thank you for your positive and constructive comments. Please see my point-by-point responses below. Any changes to the manuscript are highlighted in red color.
Point 1: Please clarify what exactly the 'double template' was in figure 3. How many copies of each bacterium?
Response 1: “Double Template” meant the RPA reactions had both V. vulnificus and V. cholerae as the templates (104 copies of each, mixed). This information has been added to the figure legend of Figure 3.
Point 2: Still in figure 3, I find it not so easy to follow what combination 1 and combination 2 represent. The information was not included in figure 3 legend either but only in the paragraph from the previous page. Providing more information either directly in the figure or in the legend would make it much easier to read.
Response 2: The figure legend of Figure 3 has been re-written to convey the required information.
Point 3: In figures 4b and 5, only one image from three independent experiments was shown. Because these images were not quantitative data, lack of statistical analysis comprise the claims regarding detecion limit (such as the lowest detection limit was 1 copy). This could be addressed two ways: 1) display all three strips side by side to show that 1 copy or 1 cfu/10g shrimp can be consistently detected; or 2) using image analysis software to do densitometry analysis of the bands and quantitatively graph the results with mean +/- sd
Response 3: The three strips are now displayed side by side in Figure 4b and Figure 5. The figure legends are modified accordingly.